# Higher Accumulation of Visceral Adipose Tissue Is an Independent Risk Factor for Hepatocellular Carcinoma among Viral Hepatitis Patients with Non-Cirrhotic Livers

**DOI:** 10.3390/cancers13235980

**Published:** 2021-11-28

**Authors:** Kenji Imai, Koji Takai, Takao Miwa, Toshihide Maeda, Tatsunori Hanai, Yohei Shirakami, Atsushi Suetsugu, Masahito Shimizu

**Affiliations:** Department of Gastroenterology/Internal Medicine, Gifu University Graduate School of Medicine, 1-1 Yanagido, Gifu 501-1194, Japan; koz@gifu-u.ac.jp (K.T.); takao.miwa0505@gmail.com (T.M.); toshi_z218@yahoo.co.jp (T.M.); hanai0606@yahoo.co.jp (T.H.); ys2443@gifu-u.ac.jp (Y.S.); asue@gifu-u.ac.jp (A.S.); shimim@gifu-u.ac.jp (M.S.)

**Keywords:** hepatocellular carcinoma, visceral adipose tissue, metabolic syndrome, hepatitis B virus, hepatitis C virus

## Abstract

**Simple Summary:**

Hepatocellular carcinoma (HCC) occurring in non-cirrhotic livers is often overlooked in clinical practice because the present HCC surveillance strategies usually focus only on patients with cirrhotic livers. This study aimed to determine the risk factors for HCC among viral hepatitis patients with non-cirrhotic livers. The findings of this study could be very useful in detecting HCC at an early stage, especially in patients with viral hepatitis who may not have developed extensive cirrhosis.

**Abstract:**

This study aimed to determine the risk factors for hepatocellular carcinoma in non-cirrhotic livers among viral hepatitis patients. A total of 333 HCC cases, including 69 hepatitis B virus (HBV)-related and 264 hepatitis C virus (HCV)-related, were divided into cirrhotic (Fibrosis-4 [FIB-4] index > 3.25) and non-cirrhotic groups (FIB-4 index ≤ 3.25). The clinical characteristics of the two groups were compared. The independent risk factors for the development of HCC were analyzed using logistic regression analysis. The patients with HBV-related HCC were significantly younger, had better Child-Pugh scores, lower FIB-4 index and Mac-2 binding protein glycosylated isomers (M2BPGi) levels, more progressive cancer stage, and higher alpha-fetoprotein (AFP) levels than those with HCV-related HCC. Diabetes mellitus and hypertension were less common in patients with HBV-related HCC. The non-cirrhotic group with HBV-related HCC had a higher visceral adipose tissue index (VATI), better Child-Pugh score, and higher hemoglobin A1c (HbA1c), whereas the one with HCV-related HCC had a higher proportion of men, higher VATI, better Child-Pugh score, higher HbA1c, and a higher prevalence of hypertension, than the corresponding cirrhotic groups. Logistic regression analyses demonstrated that age, male sex, VATI, HbA1c, the presence of hypertension, and HBV etiology were independent risk factors for HCC in a non-cirrhotic liver. A high accumulation of VAT is a risk factor for HCC in patients with non-cirrhotic livers.

## 1. Introduction

Hepatocellular carcinoma (HCC) is one of the most common malignancies worldwide; more than half a million people are diagnosed with HCC annually [1]. HCC generally develops in patients with chronic liver damage due to various causative agents, such as persistent hepatitis B virus (HBV) and hepatitis C virus (HCV) infections, alcohol consumption, obesity, and diabetes mellitus (DM)-related metabolic disorders [2]. Among these, HBV and HCV infections are the most common etiological factors worldwide and account for the majority of the incident cases of HCC (40–50%) [3]. However, in the recent years, the increased prevalence of obesity and diabetes in many parts of the world has led to an increased number of obesity-related HCC cases [4]. Approximately 20% of the cases of nonalcoholic fatty liver disease (NAFLD), hepatic manifestations of obesity, and metabolic syndromes present as nonalcoholic steatohepatitis with a risk of progression to cirrhosis and HCC [5].

Regardless of these etiologies, cirrhosis precedes the diagnosis of HCC in most individuals, but cirrhosis is not always a prerequisite of HCC development [5]. According to the guidelines of the American Association for the Study of Liver Diseases, the European Association for the Study of the Liver, and the Japan Society of Hepatology (JSH) [6,7,8], cirrhosis patients are considered at a high risk for developing HCC, and frequent HCC surveillance is recommended for these patients. However, these surveillance strategies may overlook the occurrence of HCC in patients with non-cirrhotic livers. Thus, to screen for HCC, including the cases originating in non-cirrhotic livers, the risk factors for HCC in a non-cirrhotic liver should be determined.

Several studies have revealed that obesity, DM, and related factors are risk factors for HCC. A meta-analysis revealed that DM is associated with an approximately 2.5-fold increased risk of HCC [9]. A synergistic interaction between diabetes and viral hepatitis {hazard ratio (HR), 4.8; 95% confidence interval (CI), 2.7–6.9} with respect to HCC risk was observed [10]. Patients with high levels of homeostatic model assessment of insulin resistance (HOMA-IR) and visceral adipose tissue (VAT) volume have an increased recurrence risk of HCV-related HCC after a curative treatment [11,12]. However, it is not clear whether obesity or diabetes causes HCC to develop earlier when compared to the usual clinical course of viral hepatitis in the absence of these conditions.

The aim of the present study was to determine the risk factors for HCC in non-cirrhotic livers among the viral hepatitis patients who have either hepatitis B surface antigen (HBs-Ag) or hepatitis C virus antibody (HCV-Ab). For this purpose, hepatitis virus-positive HCC patients were divided into cirrhotic or non-cirrhotic groups based on the Fibrosis-4 (FIB-4) index, a simple and noninvasive index to predict significant fibrosis in patients with liver diseases [13], and the baseline demographic and clinical characteristics of the two groups were compared to determine the possible risk factors for developing HCC in a non-cirrhotic liver.

## 2. Materials and Methods

### 2.1. Enrolled Patients and Diagnosis of HCC

We treated a total of 500 patients diagnosed with HCC in our hospital between May 2006 and December 2020. Among them, we excluded 164 patients who were HBs-Ag and HCV-Ab negative, and three who were HBs-Ag and HCV-Ab positive. The remaining 333 patients (69 with HBs-Ag positive and 264 with HCV-Ab positive) were enrolled in this study. The baseline demographic and clinical characteristics of all the enrolled patients are shown in Table 1. The diagnosis of HCC was made based on the diagnostic algorithm found in the clinical practice guidelines for HCC [6]. Typical HCC was diagnosed based on lesions that were visualized as high-attenuation areas in the arterial phase, and low-attenuation areas in the portal/equilibrium phase of dynamic computed tomography (CT) or magnetic resonance imaging (MRI) compared to the surrounding liver parenchyma. Untypical lesions were diagnosed with CT during arterial portography and hepatic arteriography, or liver biopsy, and without a definitive HCC diagnosis; the patients were followed up every three months according to the above-described guideline [6].

This was a retrospective observational study that did not require new study specimens, and instead relied only on preexisting samples or medical information. Therefore, we did not require written informed consent from patients. Instead, by disclosing the details of the study, we provided the patients with the opportunity to opt-out. The study design, including the consent procedure, was approved by the ethics committee of the Gifu University School of Medicine (ethical protocol code: 29–26).

### 2.2. Determination of Liver Cirrhosis and Extracting the Risk Factors for HCC in a Non-Cirrhotic Liver

Patients with FIB-4 index more than 3.25 were diagnosed with cirrhosis based on a previous study [13]. The FIB-4 index is derived as follows:FIB-4 = {Age (year) × AST (IU/L)}/{platelet count (10^9^/L) × ALT (IU/L)^1/2^}

Clinical characteristics, including body composition indices and tumor and DM-related factors, of the cirrhotic and non-cirrhotic groups of HBV and HCV-related HCC were compared to study the risk factors for liver carcinogenesis. Body composition indices such as skeletal muscle index (SMI), subcutaneous adipose tissue (SAT) index (SATI), and VAT index (VATI) were determined as previously described [12]. The cross-sectional areas of the muscle (cm^2^) at the L3 level of the CT images were normalized by the square of the height (m^2^) to obtain SMI (cm^2^/m^2^). Similarly, the cross-sectional areas of the SAT and VAT at the umbilical point were normalized by the square of the height to obtain the SATI and VATI, respectively. The cross-sectional areas of these tissues were measured using SYNAPSE VINCENT software (Fujifilm Medical, Tokyo, Japan). Among the clinical characteristics, the independent risk factors for HCC in a non-cirrhotic liver were analyzed using logistic regression analysis. The data and the clinical characteristics obtained just before the initial treatment of HCC were used for the analysis.

### 2.3. Statistical Analyses

Baseline characteristics were compared using Student’s *t*-test for continuous variables, or χ^2^ test for categorical variables. Logistic regression analysis was used to confirm the independent risk factors for HCC in a non-cirrhotic liver. Statistical significance was defined as *p* < 0.05. All the statistical analyses were performed using R ver. 4.0.5 (R Foundation for Statistical Computing, Vienna, Austria; http://www.R-project.org/, accessed on 28 November 2021).

## 3. Results

### 3.1. Differences in Baseline Characteristics and Laboratory Data between Patients with HBV- and HCV-Related HCC

Baseline demographic and clinical characteristics of all the enrolled patients and the HBV- and HCV-related HCC patients, individually, are shown in Table 1. The HBV-related HCC patients were significantly younger (*p* < 0.001) and had better Child-Pugh score (*p* = 0.027), FIB-4 index (*p* < 0.001), M2BPGi levels (*p* = 0.003), a more progressive cancer stage (*p* < 0.001), and higher AFP levels (*p* < 0.001) than the HCV-related HCC patients. With respect to metabolic syndrome, the frequency of complications of DM (*p* = 0.031) and hypertension (*p* = 0.038) was significantly higher in the HCV-related HCC patients.

### 3.2. Risk Factors for HCC in a Non-Cirrhotic Liver

In case of HBV-related HCC (Table 2), patients in the non-cirrhotic group had higher VATI (39.0 cm^2^/m^2^ vs. 27.5 cm^2^/m^2^, *p* = 0.018), better Child-Pugh score (5/6/7/8/9/10/11/12: 32/4/2/0/1/0/0/0 vs. 14/5/6/2/2/0/0/1, *p* = 0.019), and higher hemoglobin A1c (HbA1c) (6.0% vs. 5.4%, *p* = 0.014). In case of HCV-related HCC (Table 3), the non-cirrhotic group had a higher proportion of men (male/female: 56:7 vs. 126:75, *p* < 0.001), higher VATI (40.5 cm^2^/m^2^ vs. 33.7 cm^2^/m^2^, *p* = 0.015), better Child-Pugh score (5/6/7/8/9/10/11/12: 51/8/4/0/0/0/0/0 vs. 88/58/34/15/2/3/1/0, *p* < 0.011), higher HbA1c (6.2 vs. 5.8%, *p* = 0.006), and a higher prevalence of hypertension (yes/no) (37/26 vs. 77/124, *p* = 0.005).

Multivariate logistic regression analyses demonstrated that age (HR, 0.97; 95% CI, 0.94–0.99; *p* = 0.033), male sex (HR, 3.15; 95% CI, 1.56–6.36; *p* = 0.001), VATI (HR, 1.01; 95% CI, 1.00–1.03; *p* = 0.038), HbA1c (HR, 1.30; 95% CI, 1.01–1.67; *p* = 0.040), the presence of hypertension (HR, 2.06; 95% CI, 1.16–3.67; *p* = 0.014), and HBV etiology (HR, 4.00; 95% CI, 2.02–7.94; *p* < 0.001) were independent risk factors for HCC in non-cirrhotic livers (Table 4). From the final logistic regression model, including these six independent risk factors, we obtained the parameters used in the predictive formula for the odds ratio (OR) of HCC in a non-cirrhotic liver (Appendix A). The formula can be written as follows [14]:

OR of HCC in a non-cirrhotic liver
= −0.68 − 0.03 × {Age}+ 1.15 × {Sex [Male =1, Female =0]}+ 0.01 × {VATI (cm^2^⁄m^2^)}+ 0.26 × {HbA1c (%)}+ 0.72 × {Hypertension [yes =1, no =0]}− 1.39 × {Etiology [HBV =0, HCV =1]}


Additionally, we conducted receiver operating characteristic analysis and determined that the optimal cut-off value of the OR was 0.214. The sensitivity, specificity, and area under the curve for this cut-off value were 88.9%, 56.4%, and 0.778, respectively (Appendix A).

## 4. Discussion

HCC occurring in a non-cirrhotic liver is often overlooked in clinical practice. Therefore, the present study aimed to determine the risk factors for the development of HCC among viral hepatitis patients with non-cirrhotic livers. More than half of the patients with HBV had HCC even before they developed cirrhosis, and, in the present study, HBV-positivity was found to be the most important risk factor for the development of HCC in a non-cirrhotic liver. These findings suggest that to detect HBV-related HCC at an early stage, it is essential to screen extremely high-risk groups among patients with chronic hepatitis B who may not have developed extensive cirrhosis.

Surveillance strategies for HCC in patients with chronic hepatitis B differ from one country to another. For instance, patients with chronic hepatitis B in Japan are considered to be at a high risk for HCC, whereas in Europe, those with ≥18 score points of PAGE-E, which is determined by platelet counts, age, and sex, are considered [6,8]. However, according to the latest guidelines for HCC in the US [7], adults with cirrhosis of any etiology are considered at high risk, and those with chronic hepatitis B are excluded from the high-risk group, unlike the previous guidelines [15]. All the guidelines recommend HCC surveillance using ultrasound, with or without AFP, every six months for high-risk patients [6,7,8]; however, these surveillance strategies might be inadequate for early detection of this malignancy. If we could screen the extremely high-risk groups for HCC more precisely, according to the risk factors unrelated to the liver and the presence of cirrhosis, we could improve our HCC surveillance strategy by shortening the interval of surveillance, and/or combining CT or MRI as extra screening methods

This is the first study to provide evidence that increased level of VATI, and not BMI or SATI, is an independent risk factor for the development of HCC in non-cirrhotic livers infected with HBV or HCV. It has been reported that obesity-associated oncogenic drivers, such as adipose tissue remodeling and pro-inflammatory adipokine secretion [16,17], ectopic lipid accumulation and lipotoxicity [18], and the growth effects of insulin and insulin-like growth factors [19,20], promote hepatocarcinogenesis independently or in synergy with major liver histopathology [5]. In addition, VAT is more closely related to HCC development than SAT because VAT contains greater number of large adipocytes secreting adipokines involved in hepatocarcinogenesis as compared to SAT [21]. Clinical studies have revealed that high accumulation of VAT, but not SAT, is a risk factor for the recurrence of HCC of any etiology [12] and non-viral hepatitis [22]. These findings suggest that assessing VAT, which can be measured by CT examination commonly used for surveillance of HCC, might be useful for screening patients with high-risk of developing HCC in a non-cirrhotic liver.

This study also indicated that, in addition to increased VAT levels, other metabolic disorders such as diabetes mellitus and the presence of hypertension are independent risk factors for the development of HCC in a non-cirrhotic liver. Male sex is also associated with carcinogenesis, and these findings are consistent with those of some previous reports [10,23]. Thus, even patients with non-cirrhotic liver can be at an extremely high risk for HCC if they have one or more of the risk factors identified in this study. The optimal interval of HCC surveillance and screening methods should be determined in each case, according to the possible risk factors for HCC, including metabolic disorders, age, and sex, as well as the etiology of liver disease and the presence of cirrhosis.

This study has several limitations. Firstly, it was a retrospective, single-center study, and the sample size was comparatively small. Furthermore, we compared baseline demographic and clinical characteristics at the time of diagnosis of the initial HCC, which varied from early to advanced HCC, although there were no significant differences in the stages of cancer between the cirrhotic and non-cirrhotic groups. Secondly, we defined liver cirrhosis cases as those with an FIB-4 index > 3.25. Since the Fib-4 index was principally developed and validated in patients aged between 35–65 years of age [24], it is difficult to assess fibrosis in the elderly. In the present study, the mean age was 70.5 years, and the HBV-related HCC patients were younger than the HCV-related HCC patients and thus naturally had a better FIB-4 index. We evaluated the independent risk factors for HCC in non-cirrhotic liver patients using age-adjusted logistic regression analysis to eliminate the possible influence of age as a confounder (Appendix A), and it demonstrated a similar result to the multivariate logistic regression analysis (Table 4). However, the possibility that age may act as a confounding factor cannot be completely ruled out. Recently, the Fib-5 index, that adapts alkaline phosphatase instead of age, has been found to be superior to the Fib-4 index for evaluating fibrosis [25]. This definition of liver cirrhosis (FIB-4 index > 3.25) is now one of the most common among the ones obtained from non-invasive methods [13], but a more accurate assessment of liver fibrosis with other indices, including the Fib-5 index, is needed. The optimal cut-off value for cirrhosis, particularly in patients with chronic hepatitis B, and improved, non-invasive, methods for liver cirrhosis, other than the Fib-4 index, such as the Fib-5 index, remains controversial. To overcome these limitations, a prospective study involving a large number of cirrhotic and non-cirrhotic patients without HCC, enrolled from several centers, should be conducted in the future.

## 5. Conclusions

Higher accumulation of VAT together with factors such as male sex, presence of diabetes mellitus and hypertension, and hepatitis B virus infection are risk factors for HCC in non-cirrhotic livers. Screening of HCC in patients with a non-cirrhotic liver focusing on these risk factors may be clinically significant as the diagnosis of HCC occurring in non-cirrhotic livers can sometimes be delayed.

## Figures and Tables

**Table 1 cancers-13-05980-t001:** Baseline demographic and clinical characteristics of all the enrolled patients, and the HBV and HCV-related HCC patient groups.

Variables	All Patients (*n* = 333)	HBV-Related HCC (*n* = 69)	HCV-Related HCC (*n* = 264)	*p* Value
Age (years)	70.5 ± 10.2	61.0 ± 11.8	73.0 ± 8.1	<0.001
Sex (male/female)	236/97	54/15	182/82	0.139
BMI (kg/m^2^)	22.7 ± 3.3	23.4 ± 3.4	22.6 ± 3.3	0.079
SMI (cm^2^/m^2^)	43.3 ± 8.9	44.1 ± 10.8	43.1 ± 8.4	0.404
SATI (cm^2^/m^2^)	37.9 ± 25.2	38.5 ± 26.2	37.7 ± 24.9	0.816
VATI (cm^2^/m^2^)	34.4 ± 21.7	34.0 ± 20.2	34.5 ± 22.2	0.864
Child-Pugh score (5/6/7/8/9/10/11/12)	185/75/46/17/5/3/1/1	46/9/8/2/3/0/0/1	139/66/38/15/2/3/1/0	0.027
FIB-4 index	5.92 ± 4.58	3.38 ± 1.97	6.58 ± 4.83	<0.001
M2BPGi (C.O.I.)	4.08 ± 3.86	2.33 ± 2.80	4.65 ± 3.99	0.003
Cancer stage (I/II/III/IV)	86/116/91/40	7/24/14/24	79/92/77/16	<0.001
AFP levels (ng/mL)	8640 ± 53,349	31,609 ± 109,683	2702 ± 18,332	<0.001
PIVKA-II (mAU/mL)	13,968 ± 116,921	23,161 ± 71,920	115,823 ± 126,002	0.468
FPG (mg/dL)	109.6 ± 34.5	109.0 ± 37.3	109.8 ± 33.8	0.876
FIRI (µU/mL)	10.5 ± 8.9	10.3 ± 10.9	10.5 ± 8.3	0.914
HOMA-IR	2.89 ± 3.15	2.81 ± 3.20	2.91 ± 3.14	0.822
HbA1c (%)	5.9 ± 1.0	5.7 ± 0.9	5.9 ± 1.1	0.213
TG (mg/dL)	97.0 ± 53.6	88.7 ± 46.8	99.3 ± 55.1	0.178
DM (yes/no)	88/245	11/58	77/187	0.031
HL (yes/no)	11/322	3/66	8/256	0.704
HTN (yes/no)	134/199	20/49	114/150	0.038

Values are presented as mean ± standard deviation. HBV, hepatitis B virus; HCV, hepatitis C virus; HCC, hepatocellular carcinoma; BMI, body mass index; SMI, skeletal muscle index; SATI, subcutaneous adipose tissue index; VATI, visceral adipose tissue index; M2BPGi, Mac2 binding protein glucosylation isomer; AFP, alpha-fetoprotein; PIVKA-II, proteins induced by vitamin K absence or antagonist-II; FPG, fasting plasma glucose; FIRI, fasting immunoreactive insulin; HOMA-IR, homeostasis model assessment-insulin resistance; HbA1c, hemoglobin A1c; TG, triglyceride; DM, diabetes mellitus; HL, hyperlipidemia; HTN, hypertension.

**Table 2 cancers-13-05980-t002:** Baseline demographic and clinical characteristics of HBV-related HCC divided into cirrhotic and non-cirrhotic groups.

Variables	Non-Cirrhotic HCC Group (FIB-4 Index ≤ 3.25) (*n* = 39)	Cirrhotic HCC Group (FIB-4 Index > 3.25) (*n* = 30)	*p* Value
Age (years)	60.4 ± 12.5	61.7 ± 11.0	0.664
Sex (male/female)	31/8	23/7	0.778
BMI (kg/m^2^)	23.4 ± 2.9	23.3 ± 4.1	0.851
SMI (cm^2^/m^2^)	43.6 ± 11.1	44.7 ± 10.7	0.682
SATI (cm^2^/m^2^)	39.6 ± 19.4	37.2 ± 33.4	0.711
VATI (cm^2^/m^2^)	39.0 ± 21.4	27.5 ± 16.8	0.018
Child-Pugh score (5/6/7/8/9/10/11/12)	32/4/2/0/1/0/0/0	14/5/6/2/2/0/0/1	0.019
Cancer stage (I/II/III/IV)	4/15/6/14	3/9/8/10	0.718
AFP (ng/mL)	17,812 ± 81,842	49,085 ± 136,685	0.246
PIVKA-II (mAU/mL)	27,226 ± 88,484	18,013 ± 43,726	0.604
FPG (mg/dL)	112.8 ± 41.5	104.1 ± 31.0	0.351
HOMA-IR	2.5 ± 1.8	3.2 ± 4.3	0.425
HbA1c (%)	6.0 ± 1.0	5.4 ± 0.7	0.014
TG (mg/dL)	98.8 ± 53.6	75.9 ± 33.2	0.062
DM (yes/no)	6/33	5/25	1.000
HL (yes/no)	1/38	2/28	0.576
HTN (yes/no)	13/26	7/23	0.429

Values are presented as mean ± standard deviation. HBV, hepatitis B virus; HCC, hepatocellular carcinoma; BMI, body mass index; SMI, skeletal muscle index; SATI, subcutaneous adipose tissue index; VATI, visceral adipose tissue index; AFP, alpha-fetoprotein; PIVKA-II, proteins induced by vitamin K absence or antagonist-II; FPG, fasting plasma glucose; HOMA-IR, homeostasis model assessment-insulin resistance; HbA1c, hemoglobin A1c; TG, triglyceride; DM, diabetes mellitus; HL, hyperlipidemia; HTN, hypertension.

**Table 3 cancers-13-05980-t003:** Baseline demographic and clinical characteristics of HCV-related HCC divided into cirrhotic and non-cirrhotic groups.

Variables	Non-Cirrhotic HCC Group (FIB-4 Index ≤ 3.25) (*n* = 63)	Cirrhotic HCC Group (FIB-4 Index > 3.25) (*n* = 201)	*p* Value
Age (years)	71.3 ± 8.6	73.5 ± 7.9	0.058
Sex (male/female)	56/7	126/75	<0.001
BMI (kg/m^2^)	22.5 ± 2.5	22.6 ± 3.5	0.892
SMI (cm^2^/m^2^)	44.8 ± 7.1	42.5 ± 8.7	0.054
SATI (cm^2^/m^2^)	36.2 ± 16.7	38.2 ± 27.0	0.565
VATI (cm^2^/m^2^)	40.5 ± 20.8	33.7 ± 22.3	0.015
Child-Pugh score (5/6/7/8/9/10/11/12)	51/8/4/0/0/0/0/0	88/58/34/15/2/3/1/0	<0.001
Cancer stage (I/II/III/IV)	16/24/21/2	63/68/56/14	0.521
AFP (ng/mL)	767 ± 3609	3299 ± 20,851	0.343
PIVKA-II (mAU/mL)	1778 ± 4694	14,558 ± 143,785	0.489
FPG (mg/dL)	115.7 ± 32.3	108.0 ± 34.1	0.128
HOMA-IR	2.9 ± 3.5	2.9 ± 3.0	0.873
HbA1c (%)	6.2 ± 1.1	5.8 ± 1.1	0.006
TG (mg/dL)	111.0 ± 53.3	95.1 ± 55.3	0.062
DM (yes/no)	22/41	55/146	0.268
HL (yes/no)	4/59	4/197	0.095
HTN (yes/no)	37/26	77/124	0.005

Values are presented as mean ± standard deviation. HCV, hepatitis C virus; HCC, hepatocellular carcinoma; BMI, body mass index; SMI, skeletal muscle index; SATI, subcutaneous adipose tissue index; VATI, visceral adipose tissue index; AFP, alpha-fetoprotein; PIVKA-II, proteins induced by vitamin K absence or antagonist-II; FPG, fasting plasma glucose; HOMA-IR, homeostasis model assessment-insulin resistance; HbA1c, hemoglobin A1c; TG, triglyceride; DM, diabetes mellitus; HL, hyperlipidemia; HTN, hypertension.

**Table 4 cancers-13-05980-t004:** Univariat4e and multivariate logistic regression analyses of pathogenic risks of HBV/HCV-related HCC in a non-cirrhotic liver.

Variables	Univariate Analysis		Multivariate Analysis	
	HR (95% CI)	*p* Value	HR (95% CI)	*p* Value
Age (years)	0.96 (0.93–0.98)	<0.001	0.97 (0.94–0.99)	0.033
Sex (male vs. female)	3.19 (1.73–5.88)	<0.001	3.15 (1.56–6.36)	0.001
BMI (kg/m^2^)	1.02 (0.95–1.09)	0.620		
SMI (cm^2^/m^2^)	1.02 (0.99–1.05)	0.141		
SATI (cm^2^/m^2^)	0.99 (0.99–1.01)	0.830		
VATI (cm^2^/m^2^)	1.02 (1.01–1.03)	0.003	1.01 (1.00–1.03)	0.038
Cancer stage (I/II/III/IV)	1.21 (0.96–1.54)	0.114		
FPG (mg/dL)	1.01 (0.99–1.01)	0.102		
HOMA-IR	0.97 (0.89–1.07)	0.557		
HbA1c (%)	1.39 (1.11–1.74)	0.004	1.30 (1.01–1.67)	0.040
TG (mg/dL)	1.00 (1.00–1.01)	0.051		
DM (yes vs. no)	1.08 (0.64–1.82)	0.778		
HL (yes vs. no)	1.93 (0.58–6.49)	0.294		
HTN (yes vs. no)	1.68 (1.05–2.70)	0.031	2.06 (1.16–3.67)	0.014
Etiology (HBV vs. HCV)	4.15 (2.39–7.19)	<0.001	4.00 (2.02–7.94)	<0.001

HBV, hepatitis B virus; HCV, hepatitis C virus; HCC, hepatocellular carcinoma; HR, hazard ratio; CI, confidence interval; BMI, body mass index; SMI, skeletal muscle index; SATI, subcutaneous adipose tissue index; VATI, visceral adipose tissue index; FPG, fasting plasma glucose; HOMA-IR, homeostasis model assessment-insulin resistance; HbA1c, hemoglobin A1c; TG, triglyceride; DM, diabetes mellitus; HL, hyperlipidemia; HTN, hypertension.

## Data Availability

The data presented in this study are available upon request from the corresponding author.

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
