# Peer review of "Higher Accumulation of Visceral Adipose Tissue Is an Independent Risk Factor for Hepatocellular Carcinoma among Viral Hepatitis Patients with Non-Cirrhotic Livers"

_cancers, 2021, doi:10.3390/cancers13235980_

Round 1
Reviewer 1 Report
The authors have addressed my previous comments.
Reviewer 2 Report
I recommend accept and publish this revised manuscript.
This manuscript is a resubmission of an earlier submission. The following is a list of the peer review reports and author responses from that submission.
Round 1
Reviewer 1 Report
General comments:
This study aimed to determine the risk factors for hepatocellular carcinoma in non-cirrhotic livers among viral hepatitis patients. They reported that higher accumulation of visceral adipose tissue is an independent risk factor for HCC among viral hepatitis patients with non-cirrhotic livers.
Major comments:
- In the end of abstract, it lacks of the conclusion sentence for this study. Please add it briefly.
- Page 5: “From this logistic regression model, the predictive formula for the odds ratio (OR) of HCC in a non-cirrhotic liver can be written as follows [14]”. Please explain how to get these values for each parameter.
- Please discuss the application or suitability of receiver operating characteristic (ROC) in evaluating this study.
Minor comments:
- Abstract: Please provide the full name with the abbreviation for “FIB-4”, “
M2BPGi”, “AFP”, and “HbA1c”.
- section 2.2: 1. Please provide the full name with the abbreviation for “FIB-4”.
Reviewer 2 Report
Imai et al studied the risk factors for HCC in non-cirrhotic conditions among patients who have either HBs-Ag or HCV-Ab.
1) the authors stated that 'All the study participants provided verbal informed consent, which was considered sufficient as this study followed an observational research design that did not require new study specimens, and instead relied only on preexisting samples.' However, in my experience, written informed consents are usually required. Please provide study protocols and ethical clearance approved by the local IRB stating that verbal consents are sufficient.
2) Please provide more details regarding the HCC diagnosis criteria with references.
3) Using FIB-4 to determine liver cirrhosis is a bit tricky, as this FIB-4 system is to provide estimations non-invasively. Moreover, FIB-4 should be used with caution in patients over 65 years old, but the mean age of the study cohort is 70.5 ± 10.2. Is there imaging evidence to support this? In addition, there are reports showing that FIB-5 is better than FIB-4.
4) Please provide complete inclusion and exclusion criteria.
5) I do notice that the HBV-related HCC patients were significantly younger than HCV-related HCC patients. HCC patients had more progressive cancer stage and higher AFP levels but better Child-Pugh score and FIB-4 index. Could age here be a confounder? The authors should control for age and perform secondary analysis.